# Oxidative Stress in Arterial Hypertension (HTN): The Nuclear Factor Erythroid Factor 2-Related Factor 2 (Nrf2) Pathway, Implications and Future Perspectives

**DOI:** 10.3390/pharmaceutics14030534

**Published:** 2022-02-27

**Authors:** Daniela Maria Tanase, Alina Georgiana Apostol, Claudia Florida Costea, Claudia Cristina Tarniceriu, Ionut Tudorancea, Minela Aida Maranduca, Mariana Floria, Ionela Lacramioara Serban

**Affiliations:** 1Department of Internal Medicine, “Grigore T. Popa” University of Medicine and Pharmacy, 700115 Iasi, Romania; tanasedm@gmail.com (D.M.T.); floria_mariana@yahoo.com (M.F.); 2Internal Medicine Clinic, “St. Spiridon” County Clinical Emergency Hospital, 700115 Iasi, Romania; 3Department of Neurology, “Grigore T. Popa” University of Medicine and Pharmacy, 700115 Iasi, Romania; georgianaapostol07@gmail.com; 4Neurology Clinic, Clinical Rehabilitation Hospital, 700661 Iasi, Romania; 5Department of Ophthalmology, Faculty of Medicine, “Grigore T. Popa” University of Medicine and Pharmacy, 700115 Iasi, Romania; costea10@yahoo.com; 62nd Ophthalmology Clinic, “Prof. Dr. Nicolae Oblu” Emergency Clinical Hospital, 700309 Iasi, Romania; 7Department of Morpho-Functional Sciences I, Discipline of Anatomy, “Grigore T. Popa” University of Medicine and Pharmacy, 700115 Iasi, Romania; cristinaghib@yahoo.com; 8Hematology Clinic, “St. Spiridon” County Clinical Emergency Hospital, 700111 Iasi, Romania; 9Department of Morpho-Functional Sciences II, Discipline of Physiology, “Grigore T. Popa” University of Medicine and Pharmacy, 700115 Iasi, Romania; minela.maranduca@umfiasi.ro (M.A.M.); ionela.serban@umfiasi.ro (I.L.S.); 10Cardiology Clinic “St. Spiridon” County Clinical Emergency Hospital, 700111 Iasi, Romania; 11Internal Medicine Clinic, Emergency Military Clinical Hospital, 700483 Iasi, Romania

**Keywords:** arterial hypertension, HTN, nuclear factor erythroid factor 2-related factor 2, Nrf2, oxidative stress, antioxidant

## Abstract

Arterial hypertension (HTN) is one of the most prevalent entities globally, characterized by increased incidence and heterogeneous pathophysiology. Among possible etiologies, oxidative stress (OS) is currently extensively studied, with emerging evidence showing its involvement in endothelial dysfunction and in different cardiovascular diseases (CVD) such as HTN, as well as its potential as a therapeutic target. While there is a clear physiological equilibrium between reactive oxygen species (ROS) and antioxidants essential for many cellular functions, excessive levels of ROS lead to vascular cell impairment with decreased nitric oxide (NO) availability and vasoconstriction, which promotes HTN. On the other hand, transcription factors such as nuclear factor erythroid factor 2-related factor 2 (Nrf2) mediate antioxidant response pathways and maintain cellular reduction–oxidation homeostasis, exerting protective effects. In this review, we describe the relationship between OS and hypertension-induced endothelial dysfunction and the involvement and therapeutic potential of Nrf2 in HTN.

## 1. Introduction

Arterial hypertension (HTN) represents a persistent increase in blood pressure (BP), measuring at least 140/90 mmHg according to the European Society of Hypertension (ESH) and European Society of Cardiology (ESC), and it is one of the major risk factors for cardiovascular diseases (CVD) [1]. It is estimated that 1.13 billion people suffer from hypertension worldwide, making it the main cause of strokes and coronary heart disease (CHD). Therefore, approximately 8.5 million deaths per annum are attributed to raised BP [2]. Considering the context of the COVID-19 pandemic, it is worth mentioning that almost all available evidence up to this date suggests that HTN can be a risk factor of severe COVID-19 disease and can also become a serious sequela [3,4,5,6]. The etiology of hypertension, as it is complex and involves a myriad of varied factors, continues to remain one of the top scientific subjects of interest. While the role of inflammation and oxidative stress (OS) in HTN is currently extensively explored [7,8,9,10], the precise deleterious effect of HTN on endothelial integrity and the involvement of molecules such as the nuclear factor erythroid 2-related factor 2 (Nrf2) in the pathogenesis and evolution of raised BP remain elusive [11,12].

In this narrative review, we aim to describe the relationship between OS and hypertension-induced endothelial dysfunction and the role and therapeutic potential of Nrf2 in HTN.

## 2. Oxidative Stress in Hypertension

Oxidative stress is a process that takes place inside the cells and occurs in situations where there is an increased number of reactive oxygen species (ROS) or a low antioxidant response towards cell aggression [13,14,15]. There are several endogenous sources of ROS within cardiac myocytes, especially enzymes such as mitochondrial enzymes, nicotinamide adenine dinucleotide phosphate (NADPH) oxidase, xanthine oxidoreductase (XOR), and uncoupled endothelial nitric oxide synthase (eNOS) [16,17,18]. OS exerts its harmful effects on tissues by causing endothelial lesions and enlargement and thickening of the heart’s walls, mainly of the left ventricle, thus serving as one of the main processes that lead to HTN and other CVDs [19]. Endothelial dysfunction is characterized by decreased NO, with or without an imbalance between endothelium-derived relaxing and contracting factors combined with a prothrombotic and a proinflammatory state [20].

An immense variety of enzymes work to keep the balance inside the cells. In addition, molecules such as glutathione and ascorbate act as direct antioxidants in increased blood pressure conditions [21]. For a long time, researchers have been investigating the precise role of OS as a generator or potential therapeutic target, not only in the pathophysiology of HTN [22], or in pulmonary hypertension [23,24], but also in endothelial dysfunction associated with other cardiovascular and metabolic diseases [25]. Therefore, we will briefly discuss the main precursors of OS and their involvement in HTN.

### 2.1. ROS and Nitric Oxide

ROS following OS have extremely harmful cardiac effects because the heart is a big consumer of oxygen (~8–15 mL O_2_/min/100 g tissue while resting, and it can go up to more than >70 mL O_2_/min/100 g tissue while exercising) [26,27]. During sustained physical activity, molecular O_2_ is no longer reduced to water, but to superoxide O_2_−. This free radical is the precursor to most of the other ROS: hydrogen peroxide (H_2_O_2_), hydroxyl radical (·OH), singlet oxygen (1O_2_), and alpha-oxygen (α-O). ROS are also known to have benefits such as defense against pathogens, but increased levels can lead to cardiovascular apoptosis and ischemic injuries [28,29,30]. The main harmful processes exerted on cardiac myocytes are DNA and RNA damage, lipid peroxidation, deactivation of specific enzymes, and oxidation of amino acids [31]. Enhanced levels of ROS have been associated with HTN; however, new generations of protein targets that include ROS-forming or toxifying enzymes that could act as pharmacological agents are needed [32,33].

Nitric oxide (NO) is a colorless gas, a molecule produced by the myocardium in physiological conditions, and its major function is to dilate vessels. Decreased levels of NO are associated with cardiovascular diseases [34,35]. There are four known NOS isoforms: endothelial NOS (eNOS), found in cardiac and coronary endothelium; neuronal NOS (nNOS), found in cardiac cells; inducible NOS (iNOS), derived from neutrophils or myocytes in conditions of inflammation; and mitochondrial NOS (mtNOS), present in cardiac mitochondria [36,37]. The beneficial roles of NOS are mediating heart protection, improving endothelial integrity, and decreasing injuries of reperfusion caused by ischemia [38]. In ischemic situations, NO accumulates and produces ROS, leading to cardiac insults [36]. Usually, eNOS is the main source of endothelium-derived NO [39]. In order to synthesize NO from eNOS, a series of cofactors are involved, and its disruption leads to a monomeric form of the enzyme, which is uncoupled, and instead of NO, superoxide is formed. Uncoupled eNOS has been observed in patients with essential hypertension and atherosclerosis [40,41]. Endogenously activated Nrf2 and eNOS are thought to play a role in OS induced by myocardial infarction [36]. Higashi et al. [42] studied deficiency of tetrahydrobiopterin (BH4), a cofactor for NO synthase, and concluded that BH4 restores endothelium-dependent vasodilation in hypertensive patients.

### 2.2. Mitochondria

Mitochondria have multiple functions, being a variable source of ROS under physiological conditions and maintaining the redox status inside cells [43]. The small amount of ROS produced during respiration is not dangerous, and it is detoxified by endogenous means. However, in hypoxic or ischemic conditions, mitochondria generate elevated levels of ROS, which increase the chances of apoptosis and even myocardial infarction (MI) [44]. An increasing number of studies have emphasized that mitochondria can produce high levels of ROS in hypertensive animal models [45,46,47]. Dikalov et al. [48] have shown in their study that endothelial tissue treated with angiotensin II increased mitochondrial ROS generation and the damage caused to the cells and the body systems, such as decreased membrane potential and decreased respiratory control ratio, respectively. Aside from the peripheral effects, ROS can regulate blood pressure via central mechanisms. Mitochondrial superoxide is overproduced in conditions of activated renin–angiotensin system (RAS) in the central nervous system (CNS) [49]. More than that, the mitochondrial ROS activate Nrf2 and promote the expression of genes involved in the control of mitochondrial and antioxidant genes via various protein kinases [50]. Taking into consideration the mitochondrial implications in cardiovascular pathophysiology, further studies could lead to new treatment strategies in hypertension.

### 2.3. NADPH Oxidase (Nox)

NADPH oxidase (Nox) is a membrane-bound enzyme complex considered an important source of ROS in cardiac cells. This is the only known category of enzymes specialized in producing ROS, and it was first described in immune cells such as macrophages and neutrophils as molecules with antimicrobial properties. The Nox family is comprised of Nox 1, Nox 2, Nox 3, Nox 4, Nox 5, dual oxidase 1 (Duox 1), and Duox 2 which in states of hyperactivity produce excessive levels of ROS, contributing to endothelial dysfunction, inflammation, and cardiovascular remodeling. Angiotensin II regulates Nox function in vessels [51,52,53,54]. Mice and rats treated with this peptide hormone expressed increased generation of ROS and enhanced activity of Nox 1, Nox 2, and Nox 4 [21]. In order to establish how Nox-produced ROS influence blood pressure, many researchers conducted genetic studies. Their results noted that in mice with Nox 1 deletion, hypertension cannot be induced by angiotensin II. Overexpression of human Nox 1 in experimental animals revealed increased blood pressure and aortic superoxide production as a result of angiotensin II action in vascular smooth muscle cells, ventricular hypertrophy, and oxidative stress [55]. In fibroblast-specific deficiency of Nox 2 knockout mice, the response to angiotensin II was considerably decreased. This resulted in a decreased hypertensive response and an inhibited vascular smooth muscle growth [56]. On the other hand, Nox 4 knockout mice infused with angiotensin II showed no change in blood pressure. However, in these mice, vascular inflammation, thickening of the media, and endothelial dysfunction have been noted, showing that Nox 4 exerts beneficial effects on the cardiovascular system. The experimental animals which developed hypertension were treated with nonspecific Nox inhibitors (apocynin or diphenylene iodonium) and the specific inhibitor gp91 ds-tat. Consequently, both the blood pressure and the vascular OS in those animals were lowered [21].

As is known, the glucose metabolism is altered in hypertension. The pentose phosphate pathway (PPP), also called the phosphogluconate pathway and the hexose monophosphate (HMP) shunt, acts as an essential element of cellular metabolism. The HMP shunt pathway plays a key role in NADPH2 and in ribose-5-phosphate formation and is involved in metabolic control by interacting with other metabolic pathways such as glycolysis, gluconeogenesis, and glucuronic acid. This pathway occurs in two phases that are illustrated by many reactions. The oxidative phase reactions are catalyzed by prostaglandin (PGD) and by the glucose-6-phosphate dehydrogenase (G6PD), both known to be controlled by Nrf2 [57,58]. G6PD is involved in the pathogenesis of pulmonary artery remodeling and occlusive lesion formation within the hypertensive lungs [59]. In the nonoxidative phase, NRF2 positively regulates the expression of transaldolase 1 (TALDO1) and transketolase (TKT) [58]. As deficiency in the HMP pathway can lead to different disorders, research data suggest the HMP shunt may have potential as a therapeutic target.

### 2.4. XOR

Xanthine oxidoreductase is an important source of superoxide and hydrogen peroxide in conditions of heart ischemia, inflammation, and OS. Its catalytic properties transform hypoxanthine to the end-product uric acid [60]. Mervaala et al. [61] studied the involvement of XOR in vascular damage induced by angiotensin II using double-transgenic rats (dTGRs) harboring human renin and human angiotensin genes. They showed that these rats presented overactivity of XOR in kidneys in contrast with control rats. The activity of the enzyme was successfully decreased using valsartan, an angiotensin II type 1 receptor antagonist. Moreover, valsartan, 30 mg/kg for three weeks, reduced not only blood pressure, but also cardiac hypertrophy and 24 h proteinuria. In the same study, oxypurinol, an XOR inhibitor, was used to preincubate renal arteries, resulting in an endothelium-dependent vascular relaxation by 20%. However, it has been proven that ROS generated by XOR are not the major factors responsible for endothelial dysfunction in dTGRs and that other enzymes might play a major role in angiotensin II-induced vascular dysfunction in these rats [62].

Data so far show that OS and endothelial dysfunction are causes or consequences of HTN. Dysfunctional endothelium secondary to OS and its derivative molecules and pathways loses its capacity to protect the vessel wall, with the subsequent possibility of smooth muscle cell proliferation, monocyte adhesion, raised adhesion molecule expression, and finally development of atherosclerosis. Therefore, in order to prevent this vicious chain of events, researchers focus their attention on different pathways or molecules which contribute to OS-induced HTN.

## 3. Nrf2

Nrf2 was first discovered in 1994 by Moi et al. [63], and it is a nuclear transcription factor that plays a major role in regulating the cellular adaptive antioxidant response. It is a master regulator of cytoprotective responses [64]. Nrf2 does not have antioxidative functions but exerts antioxidant effects by activating the transcription of target antioxidant genes: HMOX-1, NQ01, MT1A, superoxide dismutase (SOD), catalase (CAT), glutathione peroxidase (GPx), glutathione-S-transferase (GST), and γ-glutamylcysteine synthase (γGCS) [65,66]. Nrf2 is a polypeptide composed of 605 amino acids and 7 domains (Neh1, Neh2, Neh3, Neh4, Neh5, Neh6, Neh7). Kelch-like ECH-associated protein 1 (Keap1) inhibits Nrf2’s transcriptional action by keeping it bound to itself under physiological conditions [67]. Keap1 is a polypeptide composed of 624 amino acid residues and 5 domains: NTR (N-terminus), IVR, BTB/POZ DGR, and CTR (C-terminus). Scientific research established in cell lines and animal models that Nrf2/Keap1 pathway activation exerts protective effects in ischemia–reperfusion injury in vessels [68].

Under oxidative stress conditions, Nrf2 can be phosphorylated by several enzymatic pathways. Protein kinase C phosphorylates Nrf2 on Ser40 and allows Nrf2 to detach from Keap1 [69]. Other kinases that can modulate Nrf2’s activity include extracellular signal-regulated kinase (ERK) [70], phosphoinositide 3-kinases (PI3K) [71], AMP-activated protein kinase (AMPK) [72], and mitogen-activated protein kinase (MAPK) [73]. Phosphorylated Nrf2 (p-Nrf2) then binds to antioxidant response elements (AREs) in the nucleus, triggering the transcription of various genes which encode antioxidants, detoxifying enzymes, proteasomes, and antiapoptotic proteins aiming to scavenge excessive ROS. Nrf2 can also be activated through the canonical mechanism, where ROS oxidize part of the cysteine residues in Keap1, which in turn decreases Nrf2 ubiquitination and increases Nrf2 translocation to the nucleus, where the antioxidant transcription process can be initiated [74,75]. More than that, in Nrf2’s activity, phosphorylation by kinases plays a vital role in its posttranslational regulation. Notably, glycogen synthase kinase-3 (GSK-3) regulates negatively, whereas AMP-activated kinase, casein kinase 2, and protein kinase C positively modulate Nrf2 activity via phosphorylation of various sites [76]. Nrf2 is the key activator of AREs [77] (Figure 1).

Although Nrf2 has been portrayed as an effective agent to protect the heart, its destructive side has been of current interest [78]. Kannan et al. [79] noted in 2013 that sustained activation of Nrf2 can also lead to cardiac dysfunction. Others have stressed the protective role of Nrf2 in the first phase of pressure overload-induced adaptation of the heart, while the Nrf2 KO showed a decreased rate of cardiac hypertrophy and recovery of cardiac function by eight weeks after transverse aortic arch constriction in rodents [80]. Considering both sides of Nrf2, it is clear that further research regarding how this transcription factor can affect the heart and blood vessels at a molecular level is needed.

### Nrf2 in Hypertension

A gamut of evidence displays the key role of Nrf2 in cardiovascular diseases [81]; in metabolic disorders [82], diabetes [83,84], or obesity [85,86,87]; and even in autoimmune, gastrointestinal, and neurodegenerative diseases or cancer [88].

Endothelial dysfunction represents a crucial step in the development of atherosclerosis. As it needs to be prevented by means of antioxidative processes, Nrf2 may represent a modality to protect cells against endogenous and exogenous oxidants and thus prevent endothelial dysfunction onset [89].

Associated with Nrf2, peroxisome proliferator-activated receptor gamma (PPARγ) is a nuclear receptor and a nutritional factor that is involved in inflammation response and homeostatic control by stimulating the expression of antioxidant genes together with the retinoid X receptor (RXR) [90]. It acts as a modulator of different pathways, such as Nrf2, RAS, and P13/Akt/NOS [91]. Blood pressure regulation, improved lipid profile and anti-inflammatory response, and ameliorated sensitivity to insulin are amongst the beneficial effects of PPARγ [92].

ROS produced during inflammation can activate Nrf2 and the nuclear factor kappa-light-chain-enhancer of activated B cells (NF-kB). Once activated, Nrf2 attenuates ROS and consequently NF-kB activity [88]. Therefore, Nrf2 and NF-kB are both transcription factors that mediate the cellular response under conditions of OS and inflammation. NF-kB plays an important role in inducing the expression of multiple proinflammatory genes, such as those encoding chemokines and cytokines [93,94]. As a consequence, disrupted NF-kB activation contributes to the pathogenesis of multiple inflammatory diseases. Bhandari et al. [95] proposed a potential interplay between Nrf2 and NF-kB, where each pathway could inhibit the transcription activity of the other factor. Moreover, this crosstalk appears to work both ways, with Nrf2 having the ability to inhibit NF-kb and the other way around. Other transcription factors that activate Nrf2 are the aryl hydrocarbon receptor (AhR), specificity protein 1 (Sp-1), myocyte-specific enhancer factor 2 D (MEF2D), p53, c-Myc, c-Jun, and breast cancer 1 (BRCA1) [78]. Additionally, recent evidence displays the major role of Nrf2 in protecting against OS and inflammation by stimulating phase II antioxidant enzymes such as glutathione S-transferase (GST), UDP-glucuronosyltransferase (UGT), UDP-glucuronic acid synthesis enzymes, and HO-1 [96,97]. Nrf2 also exerts its anti-inflammatory effect by binding to the promoter sequence of the key proinflammatory cytokines (IL-1β, IL-6) and by reducing the activity of RNA polymerase II, which will result in the suppression of gene expression [98].

The link between OS and HTN has already been proven in several animal models, but there is still room for research to also prove this in humans. Lopes et al. [99] used male Wistar Kyoto rats (WKY) and stroke-prone spontaneously hypertensive rats (SHRSP) to study how the vascular Nrf2 system influences vascular function and redox signaling. Nrf2 was downregulated in SHRSP in conditions of increased vascular OS which was connected with vascular dysfunction. However, Nrf2 activators, bardoxolone and L-sulforaphane, blocked the formation of angiotensin II-induced ROS, resulting in restored endothelial dysfunction and decreased inflammation in both WKY and SHRSP cells. Therefore, the author stresses the vasoprotective function of Nrf2 in HTN.

Another study performed by Banday and Lokhandwala [100] also investigated the role of Nrf2 in decreasing OS and BP in rats. Rats treated with L-buthionine-sulfoximine (BSO) were used to test whether sulforaphane could lower OS, decrease BP, and repair renal dopamine receptors (D1Rs). The study has successfully demonstrated the hypothesis that by activating Nrf2, the phase II antioxidant enzymes are generated, which leads to decreased OS and a well-functioning D1R system. These findings confirm that the use of sulforaphane helps in keeping BP in a normal range.

Tan et al. [69] have recently studied how the Nrf2 pathway decreases OS in the rostral ventrolateral medulla (RVLM) and whether Nrf2 intervenes in β-arrestin1’s antihypertensive action. RVLM is one of the key areas involved in the regulation of BP and sympathetic activity. β-Arrestin1 is a cytosolic protein acting as a cofactor in the desensitization of β-adrenergic receptors [101]. The results confirmed the antihypertensive role of the overexpression of β-arrestin1 in the RVLM by activating the Nrf2 pathway and increasing the generation of antioxidant enzymes. All these reactions lead to a decreased sympathetic outflow and a reduced BP value.

High BP caused by kidney diseases is often called renal HTN or renovascular HTN. It is important to mention that HTN could be both a cause and an effect of chronic kidney disease [102]. The renin–angiotensin–aldosterone (RAAS) system is responsible for regulating BP and electrolyte and fluid balance in the kidneys. The activation of this hormone system generates angiotensin II, which stimulates in turn the production of aldosterone. Aldosterone is the main mineral corticosteroid hormone, and its main function is to preserve sodium and water in the kidneys, thus causing the BP to increase [103,104,105]. Xiao et al. [106] discovered that in mice suffering from chronic heart failure which overexpressed angiotensin-converting enzyme 2 (ACE2), the sympathetic output was remarkably low. Others studied the hypothesis that ACE2 could decrease ROS generation using a pathway involving Nrf2 and antioxidant enzymes in the RVLM. The study succeeded in proving that both ACE2 and Nrf2 have the ability to inhibit sympathetic activity in HTN and chronic heart failure when they are administered in the RVLM [107]. Others noted in hypertensive rodent models that via OS and endothelial dysfunction, HTN could be one of the causes of Nrf2 transcriptional misregulations and not the other way around [99,108]. By cause of raised levels of Nrf2 repressors in hypertensive models, these results imply that the Nrf2 antioxidant defense system is insufficient to counteract the effects of OS.

Current scientific research focuses more on discovering specific factors linked to an inadequate Nrf2 signaling system, rather than on the Nrf2 antioxidant adaptative responses [108]. Nonetheless, perhaps amplifying Nrf2 activity may hold therapeutic potential for ameliorating HTN.

## 4. Therapeutic Options in HTN

The pharmacological Nrf2 activators are electrophilic compounds that can modify cysteine residues in Keap1 and ultimately its conformation through oxidation/alkylation. This process induces inhibition of Keap1-mediated degradation of Nrf2, with a consequently raised load of newly synthesized Nrf2, which augments Nrf2 transcription functions [109]. As follows, these “Nrf2 activators” actually represent Keap1 inhibitors [110].

The antioxidant defense system represented by the Nrf2 activators may include a wide range of compounds and derivatives with great potential in chronic diseases; however, few are explored in hypertension. Examples include the following: phenolic compounds (butylated hydroxyanisole, butylated hydroxytoluene, and tert-butyl hydroquinone), isopropyl sulfur cyanogen compounds (sulforaphane), 1,2-mercapto-3-sulfur ketone derivatives (oltipraz), hydrogen peroxide compounds (hydrogen peroxide, isopropyl benzene hydrogen peroxide, and 4-butyl hydroperoxide), natural compounds from plants (curcumin, resveratrol, plumbagin, tanshinone, luteolin, oleanolic acid, etc.), and compounds that are rich in arsenic, selenium, trace elements, and heavy metal ions [109,111]. Registered drugs such as statins and metformin also have the ability to activate Nrf2 [91].

### 4.1. Nrf2 Activators: Bardoxolone Methyl, Sulforaphane, and Dimethyl Fumarate

Bardoxolone methyl (BM), a semisynthetic triterpenoid, is an Nrf2 activator and NF-kB pathway inhibitor, reducing OS, inflammation, and excessive RAS activation and promoting mitochondrial functions within the cells [112,113]. It has been proven that BM can be used as a powerful antiviral treatment option in hepatocyte cultures of hepatitis B and C viruses and also in herpes simplex virus type 1 [114,115]. Due to the current pandemic situation, studies regarding the use of BM in treating SARS-CoV-2 infections are still ongoing. There are currently two clinical phase III trials ongoing that investigate the safety of bardoxolone (CDD0-ME) therapy in patients with pulmonary hypertension: the Ranger trial (NCT03068130) and the CATALYST trial (NCT02657356). As we are at the inception of exploring the Nrf2–HTN relationship, it is not unexpected that we could not identify a specific clinical trial that explores therapy with Nrf2 in HTN.

Taking this into consideration, it is necessary to understand Nrf2’s mediator role and the additional contribution of BM in the anti-inflammatory processes which take place throughout the body systems.

Sulforaphane (SFN), an isothiocyanate, is also a potent natural Nrf2 activator, which was first isolated from red cabbage and hoary cress and later from broccoli [116]. Its direct interaction with Keap1 allows Nrf2 to accumulate in the nucleus and to activate cytoprotective mechanisms in order to protect cells from ROS [117]. SFN has the ability to activate over 500 genes by means of the Nrf2/ARE signaling pathway. SFN is known as a natural compound that can induce phase II enzymes in vivo and in vitro. These enzymes play a role in inactivating and eliminating toxic substances accumulated during periods of OS within the vascular smooth muscle cells [118]. Senanayake et al. [119] studied the effect of SFN on SHRSP and Sprague Dawley (SD) rats. The research demonstrated that this Nrf2 activator decreased the three used parameters (systolic blood pressure, diastolic blood pressure, and mean arterial pressure) by 9%, 12%, and 11%, respectively, in SHRSP, while there was no effect on SD rats. Moderate activation of Nrf2 by SFN, which is basically a dietary factor, may emphasize the mild protective role of Nrf2 through a healthy diet, which could amplify the therapeutic benefits of this antioxidant-gene-modulated system.

Dimethyl fumarate (DMF) (Tecfidera), named after the earth smoke plant (Fumaria officinalis), is the methyl ester of fumaric acid. DMF has been reported to be one of the most potent Nrf2 activators, having antioxidant and anti-inflammatory properties [92]. This drug was approved in 2013 by the US Food and Drug Administration (FDA) as first-line treatment of relapsing–remitting multiple sclerosis [120] and in 2017 was approved by the European Medicines Agency (EMA) for the treatment of moderate-to-severe chronic plaque psoriasis [121].

DMF exerts potential neuroprotective, immunomodulating, antifibrotic, and radiosensitizing activities, which are dependent on Nrf2 antioxidant pathways. After oral administration, DMF is converted into its active metabolite monomethyl fumarate (MMF). MMF reacts with C151 in Keap1, thereby activating Nrf2 which subsequently translocates to the nucleus and binds to the AREs. This induces the expression of a number of cytoprotective genes, including NAD(P)H quinone oxidoreductase 1 (NQO1), sulfiredoxin 1 (Srxn1), heme oxygenase-1 (HO1, HMOX1), superoxide dismutase 1 (SOD1), gamma-glutamylcysteine synthetase (gamma-GCS), thioredoxin reductase-1 (TXNRD1), GST, and glutamate-cysteine ligase catalytic subunit (Gclc). It increases the synthesis of the antioxidant glutathione (GSH), and via inhibition of the NF-kB-mediated pathway, DMF modulates the production of certain cytokines [120]. Additionally, DMF activates the Nrf2/Keap1 pathway by S-alkylating Keap1 and by forcing BACH1, an Nrf2 inhibitor, to leave the nucleus [122].

Little attention has been given to the relation between DMF and CVDs, especially HTN. Experimental studies show that DMF reduces inflammation, oxidative damage, and fibrosis in mouse models of pulmonary arterial hypertension [123] and can attenuate aberrant remodeling after acute vascular injury in rodent carotid arteries [124]. Hsu et al. [125] studied male rats which were exposed prenatally to dexamethasone and postnatally to a diet rich in fats in order to develop HTN. After maternal DMF therapy in offspring, programmed HTN was prevented from developing. Via AKT/Nrf2 pathway, DMF reduces ROS generation and stimulates the expression of antioxidative genes regulated by Nrf2, having a protective role in pulmonary HTN or in diabetic cardiomyopathy [126]. Other researchers noted that fumaric acid and succinic acid may treat gestational hypertension by downregulating the expression of ten-eleven translocation 1 (TET1) and calcium-activated potassium channel subunit β1 (KCNMB1) [127].

Although DMF demonstrated efficacy in diverse clinical trials in other diseases, hypertension-related studies failed to show therapeutic efficacy and were dropped, terminated, or withdrawn. A double-blinded, placebo-controlled study of DMF (NCT02981082) in systemic sclerosis–pulmonary hypertension (SSc-PAH) patients was terminated due to low recruitment numbers. In a small-size pilot study, patients with pulmonary arterial hypertension-associated systemic sclerosis tolerated DMF poorly and did not provide power to suggest efficacy. Still, authors suggest that Nrf2 remains a valid therapeutic target and should be tested in future trials by using better Nrf2 agonists [128].

Considering this emerging evidence, we may consider that fumaric acid esters might be beneficial for patients with vascular diseases [129]; however, more research that could expand the treatment area of DMF to other diseases such as hypertension, with clinical utility and favorable safety profiles, is desired.

### 4.2. Natural Nrf2 Activators

A myriad of ingredients that are used daily and have the ability to promote the nuclear translocation of Nrf2 are reported in Table 1. Among these, spices were demonstrated to have potential as natural Nrf2 pathway activators that can be used in disorders caused by OS [130,131]. 

The experimental studies from Table 1 demonstrated that modulating natural products via Nrf2 pathways exerts beneficial antioxidant and anti-inflammatory effects [132,137,140] in severe heart failure [133], in radiation-induced OS [134], in induced-cardiac stress [136], in renal impairment [139], in atherosclerosis [142], and in ochratoxin-induced toxicity [143] and also shows cytoprotective and antimicrobial properties [135]; cytoprotective and cancer chemopreventive effects [138]; and anti-inflammatory, antioxidant, hepatoprotective, neuroprotective, cardioprotective, renoprotective, antiobesity, antidiabetic, and anticancer effects [141] in other chronic diseases. Although the studies mentioned did not directly explore the effect of natural products in hypertension, the administration of these natural products leads to an Nrf2 response with subsequent activation of various protective pathways which can be linked to vascular protection.

These elements bring new scientific ideas and opportunities for future researchers to explore the association between these natural Nrf2 activators and HTN and the potential therapeutic targets of these natural Nrf2 activators in this disease.

### 4.3. Other Therapeutic Options via Nrf2

Antioxidants consist of a group of enzymes and molecules that counteract the harmful effects of ROS, and their main role is to keep a balance between these two systems, not to fully reduce oxidants, as they can also protect against pathogens [144]. Antioxidants can be produced endogenously, but their level might be inadequate. Therefore, the body needs additional exogenous sources of antioxidants, usually obtained from plants or drugs [145]. Plants have been used for their antioxidant properties for a long period of time, and they have proved to be able to effectively restore the balance between oxidation and antioxidation by removing ROS. They have few side effects, decrease ROS production, and stop oxidation by restricting the beginning and spreading of redox reactions [146].

Thioredoxin (Trx), glutaredoxin (Grx), and peroxiredoxin (Prx) belong to the thioredoxin family and can be found in all organisms. They function as antioxidants and are implicated in a myriad of CVDs, including HTN, atherosclerosis, ischemic heart diseases, and cardiac hypertrophy [147]. Ahsan et al.’s review article [148] emphasized how Trx serum levels can work as biomarkers in identifying cardiac adaptative reactions. Moreover, mesenchymal stem cells genetically modified to express Trx1 showed an increased ability to proliferate and divide into cardiomyocytes, smooth muscle cells, and endothelial cells, making them a potential therapy for cardiac failure [149,150].

Inhibition of G6PD with a competitive inhibitor (6-aminonicotinamide) and with noncompetitive inhibitors (dehydroepiandrosterone and epiandrosterone) relaxes pulmonary arteries and precontracted aorta [151]. Dehydroepiandrosterone reduces pulmonary vascular resistance in pulmonary hypertensive rats [152], and researchers reported in humans that low levels of dehydroepiandrosterone sulfate are associated with increased severity of pulmonary hypertension [153]. A double-blind, randomized, placebo-controlled phase III study (NCT00581087) demonstrated that dehydroepiandrosterone improves pulmonary hypertension in chronic obstructive pulmonary disease (COPD) [154].

Additionally, researchers demonstrated that fructosamine 3-kinase (FN3K) triggers protein deglycation influencing Nrf2 activity. Sanghvi et al. [155] noted that FN3K is absent in certain cell stress such as cancer, and via Keap1, this can lead to extensively glycated Nrf2, which affects the ability of Nrf2 to bind proteins and interact with transcription cofactors. FN3K may represent a novel target of Nrf2 activity in cancer [155,156]. Even if the role of FN3K in arterial hypertension remains unclear, polymorphisms of the FN3K gene in patients with diabetes display a protective role against severe microangiopathy and macroangiopathy diabetes complications [157]. These data suggest that the FN3K–Nrf2 pathway is involved in different vascular alterations that may lead to hypertension and could represent a new therapeutic target.

Currently, multiple drugs are being used in the treatment of HTN, but they do not act via the Nrf2/Keap1 pathway. Among promising redox drugs for CVD therapy are antioxidants targeted against mitochondria (mitoQ) and Nrf2 activators (SFN, BM, DMF) [158]. There is strong evidence for resveratrol’s antioxidant and anti-inflammatory properties and its beneficial role for patients with CVD, but it was found to be less effective in obese individuals [159]. Anyway, further research into resveratrol’s properties and bioavailability is desirable.

As hypertension is a multifactorial disease, its clinical management usually includes lifestyle approaches and multiple-drug administration. So far, the efforts to find redox-based therapies are still ongoing as there is still no evidence of a fully efficient Nrf2-targeted drug in CVD, and implicitly HTN [8,109]. Further investigation and validation through clinical trials of a single Nrf2 activator via a certain pathway or a combination with another antioxidant are needed [150]. These results may provide enough scientific proof to implement a novel Nrf2–HTN therapeutic approach with practicability in clinical settings.

## 5. Conclusions and Future Directions

Arterial hypertension continues to remain an incremental and significant worldwide medical issue. The complex heterogeneity behind HTN also includes the imbalance and misregulation of OS factors. Even if HTN pathogenesis cannot be explained by only one mechanism, present-day data exhibit the deleterious effects of enhanced OS activity and excessive ROS products which perpetuate endothelial dysfunction and hypertension via different pathways. Nonetheless, research continuously brings us new molecules and transcriptional factors such as Nrf2, which in addition to its involvement in cardiovascular pathogenesis holds potential as a novel therapeutic approach in chronic diseases such as HTN. Preclinical studies support the anti-OS and anti-inflammatory effects of Nrf2 in HTN through regulation via various pathways. The pharmaceutical modulation of Nrf2 activity using natural and synthetic compounds enhances cell survival and sets in motion the endogenous defense antioxidant system. Biopharmaceutical companies are currently working on developing new drugs that target the Keap1–Nrf2 system, being challenged however by off-target effects and lack of a precise monitoring panel.

These results should encourage scientists to continue their research in this field. We hope for new discoveries which could help further demonstrate the key role of Nrf2 in HTN and its potential as a novel therapeutic target. Therefore, more research on this subject is essential for translating the OS concept into clinical practice.

## Figures and Tables

**Figure 1 pharmaceutics-14-00534-f001:**
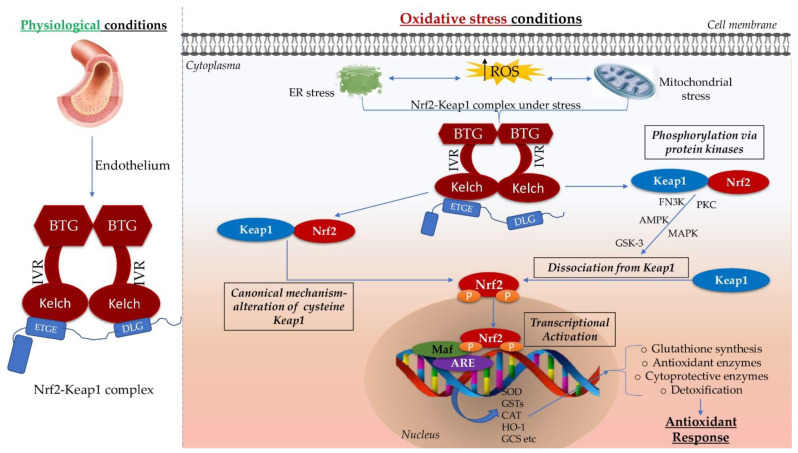
Role of the nuclear factor erythroid factor 2-related factor 2 (Nrf2) in oxidative stress. In physiological conditions, Nrf2 is bound to Keap1 (the key negative regulator and the inhibitory protein of Nrf2) and is secured to the actin cytoskeleton. This limits its transcriptional activity in the nucleus. Under OS conditions, the IVR domain leads to conformational alterations. Nrf2 is activated via canonical mechanism and/or via phosphorylation with secondary dissociation of Nrf2 from Keap1, which translocates into the nucleus and combines with the Maf protein to compose a heterodimer, capable of identifying the suitable ARE sequence. This activated ARE-mediated gene transcription is the Nrf2/Keap1–ARE pathway, which exerts antioxidant cellular functions via regulating the expression of antioxidant genes such as SOD, GST, CAT, and NQO1. Kelch-like ECH-associated protein 1 (Keap1); intervening region (IVR); endoplasmic reticulum (ER); reactive oxygen species (ROS); antioxidant response element (ARE); musculoaponeurotic fibrosarcoma (Maf); superoxide dismutase (SOD); glutathione S-transferases (GSTs), catalase (CAT); heme oxygenase-1 (HO-1); glutamylcysteine synthetase (GCS); protein kinase C (PKC); fructosamine-3-kinase (FN3K); AMP-activated protein kinase (AMPK); mitogen-activated protein kinase (MAPK).

**Table 1 pharmaceutics-14-00534-t001:** Salient effects of natural Nrf2 activators. Kelch-like ECH-associated protein 1 (Keap1); nuclear factor erythroid factor 2-related factor 2 (Nrf2); glycogen synthase kinase-3 (GSK-3); malondialdehyde (MDA); superoxide dismutase (SOD); nuclear factor kappa-light-chain-enhancer of activated B cells (NF-κB); heme oxygenase-1 (HO-1); plasma glutathione peroxidase (GSH-Px).

Authors and Ref.	Natural Compound	Organic Compound	Species and/or Cells Researched	Meaningful Findings
Kim et al. [132]	Pepper	Methysticin	Murine cell cultures murine RAW 264.7 cell line	- Oxidation or alkylation of the Keap1 proteins; - Inhibited binding of Nrf2 to Keap1; - Phosphorylation of Nrf2 by GSK-3 and subsequent proteasomal degradation.
Wafi et al. [133]	Turmeric	Curcumin	Sixty male C57BL/6 mice 10 weeks of age	- Decreased MDA and SOD levels; - Suppression of the Bax/Bcl-2-caspase-3 pathway-mediated cell death; - Diminished inflammation, fibrosis, and hypertrophy.
Ji et al. [134]	Ginger	6-Dehydrogingerdione	Human mesenchymal stem cells	- Inhibition of NF-κB activation.
Mimura et al. [135]	Rosemary	Carnosic acid	U373MG cells (human glioblastoma astrocytoma cells)	- Keap1 inactivation.
Mohan Manu et al. [136]	Water hyssop (Bacopa monnieri)	Dammarane-type triterpenoid saponins	Adult male Wistar rats	- Restored expression of Nrf2, NQO1 gene, and HO-1 followed by increased antioxidant enzymes and total glutathione levels.
He et al. [137]	Thyme	Thymol	Zebrafish	- Activated Nrf2/Keap1 pathway (signifcant downregulation of Keap1 expression and upregulation of Nrf2 expression).
Korenori et al. [138]	Wasabi	Allyl isothiocyanate	HepG2 (human hepatoma)	- Increased Keap1 modification and diminished Nrf2 degradation.
Kanlaya et al. [139]	Green tea	Catechins	Madin–Darby Canine Kidney (MDCK) renal tubular cells	- Increased antioxidative activity of phase II enzymes.
Paul et al. [140]	Ashwagandha	Triterpene lactones	Coronary artery occlusion in rats; myocardial infarction in rats	- Abrogated apoptosis in an Nrf2-dependent manner; - Increased phase II detoxification enzymes.
Farkhondeh et al. [141]	Grapes, berries, cranberries, nuts, cocoa, and dark chocolate	Resveratrol	Adult male Sprague-Dawley rats	- Significant increase in GSH-Px and SOD.
Yang et al. [142]	Tomatoes, watermelons, red carrots, grapefruits, and papayas	Lycopene	Human umbilical vein cell line	- Inhibited NF-κB nuclear translocation and transactivation.
Ramyaa et al. [143]	Apples, citrus fruits, onions, green leafy vegetables, honey	Quercetin	Vero cells (African green monkey kidney epithelial cells)	- Nuclear Nrf2 translocation.

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
