# Peer review of "Oxidative Stress in Arterial Hypertension (HTN): The Nuclear Factor Erythroid Factor 2-Related Factor 2 (Nrf2) Pathway, Implications and Future Perspectives"

_pharmaceutics, 2022, doi:10.3390/pharmaceutics14030534_

Round 1

Reviewer 1 Report

In the manuscript entitled “Oxidative stress in arterial hypertension (HTN): The nuclear factor erythroid factor 2-related factor 2 (Nrf 2) pathway, implications and future perspectives” by Tanase, D.M. et al., have reviewed various studies to describe the relation between oxidative stress and hypertension induced-endothelial dysfunction, involvement of Nrf2 and its therapeutic potential for treating hypertension. Even though the authors have reviewed the published data, they failed to discuss some of the recent studies describing the novel mechanism(s) of regulation of Nrf2 and the role these mechanisms in disease development and treatment. For instance, a study by Sanghvi et al., showed that Nrf2 activity depends on the deglycation by frucosamine kinase (FN3K). Targeted inhibition of FN3K using pharmacological agents such as DMF (1-deoxy-1-morpholinofructose), a competitive inhibitor of FN3K, is known to inactivate Nrf2 by keeping it in its glycated state. Although authors have mentioned about DMF its mechanism of action is not discussed.

The Nrf2 signaling and the role of natural Nrf2 activators is well reported in the literature. Authors would have focused more on discussing about the clinical viability of Nrf2 activators and mentioned about novel strategies to improve clinical efficacy. Further more, emphasis would have also been given about the role of Nrf2 in regulating pathways such as hexose monophosphate shunt pathway (HMP), which is a key producer of NADPH in the cells. Since NADPH is primarily involved in producing reducing compounds such as glutathione, emphasis would have been given for this pathway.

Further, the authors would have covered the pharmaceutical formulations developed using Nrf2 activators and their current status in controlling hypertension. In summary, the review is very preliminary and has not covered most recent aspects of Nrf2 and its role in hypertension

Author Response

Dear reviewer,

Firstly, thank you on behalf of our team for your time on peer-reviewing our manuscript. As fallows we have taken step by step, and tried as much as possible to correct and improve our paper according to your recommendation and observations.

-Indeed, there were some missing data that we have not included regarding some of the Nrf2 mechanism, thus we have introduced as suggested new data about dimethyl fumarate (DMF) mechanism, and scientific research that demonstrated its actions and effects in hypertension and HTN related disease. Additionally, we considered appropriate to include information about the Fructosamine- 3-kinase (FN3K)—a kinase triggers protein de-glycation in this subsection 4.3 Other therapeutic options via Nrf2, as authors such as Sanghvi el al. demonstrated that the FN3K-Nrf2 pathway is involved in different vascular alteration that may lead to hypertension and could represent a new therapeutic target.

-We included as suggested information about the importance of the glucose metabolism in HTN, with reference to the pentose phosphate pathway (PPP)/hexose monophosphate (HMP) shunt in 2.3. NADPH oxidase (Nox).

 -Although there are currently pharmaceutical formulation using Nrf2 activators in clinical trials, there are mainly studied in cancer or other chronic diseases. Our focus of research was arterial hypertension, and thus in our research we have identified few studies that describe the effect of some Nrf2 activators on hypertension per se. More, we have introduced in text additional newer research (animal studies and human studies) that shows the potential of modulating Nrf2 activity in hypertension and blood pressure control, and revised and completed with our opinion regarding the utility and viability of Nrf2 activators in hypertension.

Thank once again for your all your suggestions, we tried to revise, correct, restructure and correct any mistakes and/or grammatical issue, and ultimately improve our paper.

Reviewer 2 Report

The publication by Tanase et al. reviews the role of oxidative stress, and more particularly, Nrf2 in arterial hypertension. In general, the work is well structured and includes relevant references about this topic. However, there are some concerns that need to be fixed before acceptance:

  1. In section 2, line 66, the authors mention that eNOS is a source of ROS. However, in the next subsections 2.1 and 2.3, ROS and NO are presented separately. The first one refers to superoxide, hydroxyl radical, … but it does not mention NO, which is presented in a separated section that does not mention that this molecule is a ROS. This division is confusing.
  2. Section 2.3 is entitled NAPDH, but it refers to NOX. The title should be changed.
  3. In section 3, the authors state: “Under oxidative stress conditions, protein kinase C phosphorylates Nrf2 on Ser40 and allows Nrf2 to detach from Keap-1”. Protein kinase C is only one of the described activators of Nrf2, there are several enzymes involved such as ERK, AMPK,…. Moreover, ROS itself activate Nrf2 by interacting with the cysteine residues of Keap-1 (canonical activation) (doi:10.1016/j.phrs.2018.06.013, doi:10.1128/MCB.00099-20).
  4. Section 3 does not mention the antioxidant genes that are activated by Nrf2, responsible for the antioxidant properties of the transcription factor (SOD, CAT, GPx, …)
  5. There is a mention in lines 225-227 about the crosstalk among Nrf2 and NFkB. This point should be extended, since table 1 presents several Nrf2 natural activators and highlights their ability to inhibit NFkB (lycopene, 6-dehydrogingerdione). Also, BM and DMF have anti-inflammatory properties, as mentioned in section 4.1. This activity of Nrf2 is not only related to ROS reduction, the transcription factor also represses the expression of pro-inflammatory genes and potentiates the anti-inflammatory signalling (doi:10.1016/j.bbadis.2016.11.005)
  6. Section 4.3 is entitled “Other Nrf2 activators”. The title does not match with the content, only the last paragraph talks about Nrf2 activators, but it includes SFN, BM, DMF and resveratrol, which are mentioned in the previous sections. Why do the authors explain the enzymatic and non-enzymatic antioxidant endogenous systems in this section?
  7. The conclusion does not include any sentence about the importance of Nrf2 in HTN, or the potential use of Nrf2 activators for hypertension treatment, which are the main topics of the publication
  8. There are several grammar and typographic errors throughout the manuscript:

-Abstract: “with new evidence emerging showing”; transcription factors such as nuclear factor erythroid factor 2-related factor 2 (Nrf 2) mediates”

- Section 2.1: O2, O2-, H2O2

-Nrf2 is written several times as Nrf 2

- Section 2.3: ehanced activity; “genetic many researchers conducted genetic studies”

-Section 2.5: “a series of cofactors is involved”

-Section 3: “He is a master regulator of cytoprotective responses” instead of it is …

- Section 4.3: There is a problem with abbreviations.

- Reference 75 should be corrected

Author Response

Dear reviewer,

Thank you on behalf of our team for your time on peer-reviewing. We have taken into account your recommendations regarding our paper. Therefore, we revised and corrected some of the sections, so that it would be easier for our readers to understand the topic.

  1. We agree that the previous division in section 2 was confusing and we decided to reorganize the subsections in order to easily understand that NO is a ROS. The rest of the subsections describe the other sources of ROS.
  2. Thank you for pointing out that the title of 2.3 section was not accurate. We changed it from NADPH to NADPH oxidase (Nox).
  3. We introduced in section 3 other enzymes involved in Nrf2 activation and we also added information about the canonical mechanism of Nrf2 activation as you have recommended with corresponding bibliography.
  4. Section 3 has been improved by adding the antioxidant genes which are activated by Nrf2, and are also mentioned in Figure 1.
  5. As you have suggested, we extended the topic about the Nrf2 and NF-kB crosstalk and how Nrf2 represses the expression of pro-inflammatory genes.
  6. We agree that we shouldn’t had explained the enzymatic and non-enzymatic antioxidant endogenous systems in section 4.5, hence we deleted it. We also changed the title to Other therapeutic options via Nrf2, and completed this section with newer data in order to match with the content of the paragraphs.
  7. We have taken into account your observations regarding our conclusion, thus we edited it to emphasize Nrf2’s benefits in the treatment of HTN. We mentioned as well the importance of using the natural and synthetic activators in Nrf2 modulation and our future perspective in this field.
  8. Thank you for highlighting the grammar and typographic errors which we hope we have managed to fix as correctly as possible throughout the whole manuscript. 

-We have corrected in our abstract the phrase mentioned as it was incorrectly written;

-We corrected at Section 2.1, as pointed, please check;

-From Nrf 2, we corrected to Nrf2;

-Thank you for this observation, we have corrected what was mentioned at Section 2.3, Section 3, Section 4.3.

-Reference 75 was corrected.

Reviewer 3 Report

General comments:

The authors describe the relation between OS and hypertension induced-endothelial dysfunction, involvement of Nrf2 in HTN and its therapeutic potential.

Major comments:

  1. Table 1: The authors summarized several nrf-modulating natural products. However, the evidence or studies for these products applying in hypertension treatment was not detailed discussed. If possible, please list some literature using these products for hypertension study.

Minor comments:

  1. Line 86-88: “2” should be subscript
  2. Figure 1 is not very clear.

Author Response

Dear reviewer,

Firstly, thank you on behalf of our team for your time on peer-reviewing our manuscript. We tried to fallow your recommendation, revise and correct by section, so that it would be easier for our readers to understand the topic.

Major,

  1. As suggested, we have included a paragraph in 2 Natural Nrf2 activators, regarding your observation about Table 1, and our view on these reults. “Although these experimental studies demonstrated that modulating natural products via Nrf2 pathways their exerts their beneficial antioxidant and anti-inflammatory effects [132,137,140], in severe heart failure [133], in radiation induced-OS [134], in induced-cardiac stress [136], in renal impairment [139], in atherosclerosis [142], and in ochratoxin-induced toxicity [143]; and also, show their cytoprotective and an-ti-microbial properties [135], cytoprotective and cancer chemopreventive effects [138], anti-inflammatory, antioxidant, hepatoprotective, neuroprotective, cardioprotective, renoprotective, anti-obesity, anti-diabetic, and anti-cancer effects [141] in other chronic disease. Although the studies mentioned did not explore directly the effect of natural products in hypertension, as seen their administration lead to a Nrf2 response with subsequently activation of various protective pathways which can be linked to a vascular protection. These elements, brings new scientific ideas and opportunities to for future re-searchers to explore the association between these natural Nr2 activators and HTN, and their potential therapeutic target in this disease.”

Minor,

  1. We have corrected, thank you for your observation.
  2. We have introduced below Figure 1 the description of the processes involving Nrf2, for a better understanding and a clearer view on the antioxidant system activation in oxidative stress conditions, “In physiological conditions, Nrf2 is bound to Keap1 (the key negative regulator and the inhibitory protein of Nrf2), and is secured to the actin cytoskeleton. These limits its transcriptional activity in the nucleus. Under OS conditions, the IVR domain leads to conformational alterations with secondary dissociation of Nrf2 from Keap1, which translocates into the nucleus and combines with the Maf protein to compose a heterodimer, capable of identifying the suitable ARE sequence. This activated ARE- mediated gene transcription is the Nrf2/Keap1–ARE pathway, which via regulating antioxidant gene expression such as SOD, GST, CAT, NQO1 etc., exerts anti-oxidant cellular functions.”

Thank once again for your all your suggestions, we tried to revise, correct, restructure and correct any mistakes and/or grammatical issue, and ultimately improve our paper.

Round 2

Reviewer 1 Report

Authors have addressed the comments to the satisfaction

In the Figure 1, the authors should include the FN3K and other proteins involved in Nrf2 regulation

Author Response

Dear reviewer,

Thanks again on behalf of our team for your time and the recommendations you’d given.

We took into consideration your suggestion of enhancing our figure by adding the proteins which are involved in Nrf2’s regulation. As it can be observed, the figure is now more complex and it includes more details on how Nrf2 is dissociated from Keap1 and phosphorylated via protein kinases. Also, we have included the data mentioned in Figure 1. description. We really hope we managed to make the figure look clear and explicit and easy for the readers to understand the depicted information.

Reviewer 2 Report

The authors have answered all the revisions required

Author Response

Dear reviewer,

Thanks again on behalf of our team for your time and the recommendations you’d given.
